| Open Peer Review | Virology | Methods and Protocols

# Development and characterization of a double-fluorescent HIV-1 reporter cellular model to tackle the Rev-dependent export pathway

Jakub Wadas,[1,2] Haider Ali,[1,2] Aleksandra Osiecka,[1] Agnieszka Dorman,[1,2] Krzysztof Pyrc,[3] Anna Kula-Pacurar[1]

**ABSTRACT** The Rev-dependent nuclear export of unspliced and singly-spliced transcripts of human immunodeficiency virus type 1 (HIV-1) constitutes a critical yet poorly characterized post-transcriptional event essential for effective viral replication. In this study, we engineered a dual-fluorescent HIV-1-based cellular reporter system to elucidate the mechanisms underpinning Rev-dependent export. By generating multiple stably integrated inducible cellular clones, we ensured the expression of two distinct fluorescent proteins, mKO2, and ECFP, from unspliced (Rev dependent) and multiply spliced (Rev independent) HIV-1 transcripts, respectively. Utilizing flow cytometry, we performed quantitative analyses of dual-fluorescent cell populations. The developed tool enables precise assessment of the Rev-dependent export, and we validated it using known inhibitors of this pathway (leptomycin D), as well as targeted depletion of MATR3, an essential cofactor of Rev, and CRNKL1, a repressor of unspliced HIV-1 RNA export.

**IMPORTANCE** The developed dual-fluorescent reporter system represents a powerful and handy tool for the identification and characterization of novel molecular players involved in the Rev-dependent export pathway. This system not only holds promise for advancing our understanding of human immunodeficiency virus type 1 (HIV-1) biology but also serves as an invaluable platform for high-throughput drug screening aimed at targeting post-transcriptional HIV-1 RNA processes, particularly nuclear export. Consequently, this study offers significant implications for the development of novel therapeutic strategies to eradicate the virus.

**KEYWORDS** HIV-1, Rev protein, Rev cofactors, HIV-1 RNA processing, Rev-dependent export, reporter system, flow cytometry

Human immunodeficiency virus type 1 (HIV-1) expresses three size classes of RNAs: intron-containing unspliced (US), singly spliced (SS) RNAs, and intronless multiply spliced (MS) RNAs. MS HIV-1 RNA is translated into two regulatory proteins, Tat and Rev, and accessory protein Nef. Other accessory proteins, Vif, Vpr, and Vpu are translated from SS HIV-1 RNA that also codes for structural proteins gp120 and gp41, localizing to the surface of the viral envelope. The US HIV-1 RNA represents a full-length genomic RNA, which is packed into the newly formed virions. It also serves as a template for the translation of Gag and Pol proteins. Nuclear export of these three size classes of HIV-1 RNA is a complex process. MS HIV-1 RNA leaves the nucleus by the classical nuclear export Tap/NXF1-dependent pathway used by cellular mRNAs. However, US and SS HIV-1 RNA transcripts require an alternative mechanism of nuclear export utilizing the Rev viral protein (1). Several pioneering studies have unveiled the molecular details of Rev's function. Rev specifically binds to the Rev-responsive element (RRE) region, a structure located within the env sequence of US and SS HIV transcripts (2, 3). Once a sufficient

**Peer Reviewer** Sebla Bulent Kutluay, Sebla Bulent Kutluay, Washington University in St Louis School of Medicine, Saint Louis, Missouri, USA

Address correspondence to Anna Kula-Pacurar, anna.kula-pacurar@uj.edu.pl.

The authors declare no conflict of interest.

See the funding table on p. 15.

concentration of Rev is reached in the nucleus, multiple Rev molecules multimerize and bind to the RRE (4). Upon binding, Rev facilitates the nuclear export of these transcripts through the nuclear pore complex (NPC) by utilizing the chromosomal region maintenance 1 (CRM-1) pathway (5, 6). Interestingly, recent function of Rev was revealed to suppress the Tap/NXF1 binding to RRE-containing HIV-1 RNAs by competing with the cap-binding complex, thereby inhibiting the recruitment of TREX complex to viral US and SS RNAs, which may explain the ability of Rev to utilize the above-mentioned non-canonical nuclear export pathway via CRM-1 (7).

The Rev interactome has been extensively studied, revealing various cellular cofactors that contribute to Rev's function in regulating viral RNA. The most known Rev cofactors are members of DEAD (D-E-A-D: Asp-Glu-Ala-Asp)-box RNA helicase family, which is a large group of conserved RNA-binding proteins involved in RNA metabolism, including splicing, translation, ribosome biogenesis, and RNA decay (8). Their helicase activity allows for unwinding RNA helices to facilitate RNA-protein interactions (9). A DEAD-box family member-1, DDX1, promotes the export of RRE-containing RNAs (10). More specifically, it was shown that DDX1 is a helicase exhibiting RNA-dependent ATPase activity and acts as an RNA chaperone, facilitating the conformational changes within the RRE region to accelerate the binding of the first Rev monomer during the nucleation step of Rev-RRE assembly. Another protein from the DEAD-box family is an ATP-dependent helicase DDX3 that binds the HIV-1 RNA/Rev/CRM-1 export complex and associates with NPCs. DDX3 induces conformational changes within RRE, facilitating the translocation of RRE-containing RNAs through the NPCs (11). Moreover, other helicases from the DEAD-box family, such as DDX5, DDX17, DDX21, DDX56, and DDX24, were also demonstrated to enhance the nuclear export of RRE-containing RNAs (12). Another cellular factor positively acting on the nuclear export by interacting with the RNA-Rev-CRM1 complex is the Src-associated protein of 68 kDa in mitosis (Sam68). Sam68 binds Rev in the HIV-1 RNA/Rev/CRM-1 complex, directing it and docking it to the NPC for subsequent nuclear export (13). Additionally, it was shown that Sam68, by binding to RRE independently from Rev, supports Tap/NXF1 in exporting RRE-containing RNAs (14). Cellular factor, human Rev-interacting protein (hRIP), also known as Rev/Rex activation domain-binding protein (Rab) also cooperates with HIV-1 RNA/Rev/CRM-1 complex by interacting with NES domain on Rev. hRIP was identified to affect the distribution of incompletely spliced RNAs by promoting the release of such RNAs from the nuclear periphery and directing them to nuclear export (15, 16). There are also other factors that increase Rev-dependent export by binding to Rev, including Staufen-2 and eukaryotic initiation factor-5A (eIF-5A) (17, 18). On the other hand, Rev activity may be hampered by the recently identified Crooked neck pre-mRNA splicing factor 1 (CRNKL1) that was shown to bind US HIV-1 RNA and induce its nuclear retention (19). Moreover, nuclear factor 90 (NF90ctv) was found to interact with RRE and compete with Rev for binding to US and SS HIV-1 RNA (20). In addition, we and others have previously identified two novel Rev cofactors: the nuclear matrix-associated RNA-binding protein Matrin 3 (MATR3) and the polypyrimidine tract-binding protein-associated splicing factor (PSF, also known as SFPQ), as nuclear factors that commit the unspliced HIV-1 RNA to Rev for export (21–23). Interestingly, recently, we demonstrated the relevance of these factors in latency and reactivation from latency (24). More specifically, we showed that low levels of MATR3 and PSF in *ex vivo* cultures of CD8$^+$-depleted peripheral blood mononuclear cells from antiretroviral therapy (ART)-treated HIV$^+$ patients correlated with incapacities of certain latency-reversing agents (LRAs) of "shock-and-kill" cure therapy. This work highlighted that low levels of MATR3 and PSF in resting cells may represent a novel post-transcriptional block in latency linked with nuclear RNA export (24). Moreover, a previous study by the Yukl group (25) has revealed several additional blocks to transcriptional elongation, polyadenylation, and splicing, suggesting that latency can also be maintained by less-characterized mechanisms operating at the post-transcriptional level. Recently, we identified a novel post-transcriptional latency block to Rev-dependent export linked to limiting levels of MATR3 causing nuclear retention of unspliced HIV-1

transcripts in *ex vivo* cultures of ART-treated PLWH (people living with HIV) (26). Hence, the development of proper tools for studying the Rev-dependent export is crucial not only for the identification and characterization of novel Rev-interacting factors but also for dissecting the role of less characterized post-transcriptional stages of HIV-1 RNA metabolism in latency and reactivation with the goal of identifying potential strategies to control or eradicate the latent virus, the last obstacle to reach an HIV/AIDS cure.

In this work, we have developed a convenient model for the identification of novel cellular factors involved in HIV-1 export and for future screenings in search of drugs tackling post-transcriptional stages. For this, we have developed a dual fluorescent reporter cellular system to tackle HIV-1 RNA processing, especially the Rev-dependent export of HIV-1 RNA. We generated stable cell lines containing silent, integrated provirus that were activated by Tat and Rev, and flow cytometry analysis of the double fluorescent mKO2 + ECFP + cell population allowed us to calculate the Rev efficiency.

## MATERIALS AND METHODS

### Cell cultures and plasmids

HEK 293T cells (human embryonic kidney 293 cells, ATCC: CRL-3216) and U2OS cells (U2 osteosarcoma, ATCC: HTB-96) and cell lines derived therefrom were cultured in DMEM culture medium supplemented with 5% (D5) and 10% (D10) FBS, respectively, with the addition of penicillin (100 U/mL) and streptomycin (100 µg/mL). Cells were cultured in 75 $cm^2$ culture vessels at 37°C, 5% $CO_2$, and 95% humidity. The culture medium was changed three times during the week with cells splitting.

Plasmids for lentiviral vector productions (psPAX2 [#12260] and pMD2G [#12259]) were obtained from the Addgene repository. pHIV-HY, pHIV-Intro, pTat-101, pRev-GFP, shLuc, and shMATR3 (TRCN0000074906) were a kind gift from Alessandro Marcello lab (ORCID 0000-0002-8903-8202). Lentiviral vector encoding Tat-IRES-Rev transgene was designed and obtained from VectorBuilder (pLV-Tat-IRES-Rev-Blast; Vector ID: VB240601-1127pfx).

### Antibodies, immunofluorescence, and western blot

For the detection of endogenous MATR3 protein, an indirect immunofluorescence staining was performed as follows: fixed samples prior to flow cytometry analysis were permeabilized for 5 min using 0.1% Triton-X 100 diluted in phosphate-buffered saline (PBS), followed by three washes in PBS. Next, cells were blocked in 5% bovine serum albumin in PBST (0.1% Tween-20 in PBS) overnight at 4°C; thereafter, cells were incubated with primary antibodies against MATR3 (Bethyl Laboratories, A300-591A, 1:500) for 2 h at room temperature. In the next step, cells were washed thrice using PBST buffer, followed by a 2-h incubation with fluorescently labeled Atto 633 fluorophore secondary antibodies (Sigma-Aldrich, 41176, 1:5,000) at room temperature. In the final step, cells were washed and analyzed by flow cytometry. Immunoblots were performed as described before (21) with the following antibodies: mKO2 (Medical and Biological Laboratories, PM051M, 1:500), GFP (St. John's Laboratory, STJ140006, 1:1,000), MATR3 (Sigma-Aldrich, MABN1587, 1:1,000), CRNKL1 (NovoPro Bioscience, #175347-50, 1:1,000), GAPDH (Cell Signaling Technology, 14C10, 1:4,000), anti-rabbit-HRP (Sigma-Aldrich, A0545, 1:20,000), and anti-mouse-HRP (Dako, P0447, 1:20,000).

### Construction of pHIV-dual plasmid

For the construction of the pHIV-dual plasmid, we took advantage of a previously described and characterized HIV-based plasmid named pHIV-intro (22). Briefly, the plasmid has been linearized by restriction enzymes SpeI (Thermo Scientific, ER1252) and BstBI (Thermo Scientific, ER0121) digestion at sites located behind the truncated *gag* gene. Next, a PCR-amplified *mKO2* gene with inserted restriction sites (primers listed in Fig. S1) was ligated at a ratio of 1:5 (linearized plasmid to PCR product) overnight at 4°C

followed by transformation to Stbl3 bacteria. From the obtained bacteria, plasmids were isolated and sequenced for properly integrated *mKO2* gene.

## RNA isolation and RT-qPCR

For reverse transcription, PCR, and quantitative PCR (qPCR) analysis, the RNA was extracted from total cell extracts using Trizol (Invitrogen) according to the manufacturer's protocol. RNA was treated with DNase I (Invitrogen) to remove genomic DNA contamination and used as a template to synthesize cDNA using random hexamers and M-MLV reverse transcriptase (Invitrogen) according to the manufacturer's protocol. Endpoint RT-PCR (35 cycles, annealing temperature: 60°C) was conducted with specific primers A + B (unspliced HIV RNA), A + C (spliced HIV RNA), TAR (total HIV RNA), and GAPDH (RNA loading control), which are shown in Fig. S1. Real-time PCR amplification was conducted in the presence of GoTaq qPCR SYBR mix (Promega) and monitored on CFX96 Thermal Cycler (Bio-Rad). The viral RNA was quantified relative to the GAPDH mRNA expression and is shown as a fold change in comparison to control samples. Results were expressed as mean ± SD. The Student's *t* test was used to test the statistical significance, and values <0.05 were considered significant.

## Lentiviral vector production, titration, and transduction

Vesicular stomatitis virus glycoprotein G (VSV-G)-pseudotyped lentiviral vectors were produced in HEK 293T cells by co-transfecting the pHIV_dual plasmid with the HIV packaging plasmid (psPAX2) and the VSV-G expression (pMD2G) plasmids. Plasmids were transfected using the PEI protocol. The medium was refreshed 16 h after PEI transfection. The lentiviral vector-containing supernatants were collected 48- and 72-h post-transfection, clarified by centrifugation, filtered through a 0.45 µm syringe filter, concentrated, aliquoted, and further stored at −80°C until further use.

For HIV-dual lentiviruses, titration was performed in a 6-well format by adding four dilutions of vectors onto the cells. The medium was refreshed after 20 h to remove unbound lentiviral particles and subjected to activation by Tat transfection and PMA + Ionomycin and TNFα stimulation for 24 h. The titer of vectors was calculated by flow cytometry measuring %ECFP$^+$ population and using the equation: TU [1/µL] = (%ECFP$^+$ × $C$ × $D$)/($V_{medium}$ + $V_{lenti}$), where TU, transductive unit; $C$, density of seeded cells; $D$, dilution factor of lentiviral vectors; $V_{medium}$, volume of medium (µL); and $V_{lenti}$, volume of added lentiviral vectors (µL).

## Fluorescent-activated cell sorting

U2OS cells transduced with HIV_dual lentiviral vector were trypsinized and washed twice with 1× PBS buffer without calcium and magnesium. The cells were then transferred to a sterile 1.5 mL polypropylene tube through a 40 µm pore size cell strainer (Corning). The cell mixture prepared in this way was placed in a fluorescent-activated cell sorting (FACS) machine MoFlo XDP (Beckman Coulter). Individual cells were then sorted into wells of a 96-well plate containing 100 µL culture medium each. The plates, after sorting, were placed in an incubator for 2 weeks until the formation of visible cell colonies.

## Flow cytometry analysis

Cells were washed twice using 1× PBS and trypsinized. Next, cells were spun down and washed with 1× PBS and then fixed using 3.7% paraformaldehyde (PFA, Sigma-Aldrich) in PBS for 10 min at room temperature. Cells were washed three times using 1× PBS and transferred into polystyrene tubes (Corning). Samples were analyzed using BD LSRFortessa Cell Analyzer (BD Biosciences). To quantify Rev efficiency, we employed the gating strategy illustrated in Fig. S2. Efficiency was calculated using the following equation: (% mKO2$^+$high ECFP$^+$)/(% total ECFP$^+$) × 100.

## Lipofectamine 2000 transfections

Transfections were performed using Lipofectamine 2000 Reagent (ThermoScientific) following the manufacturer's protocol, in a ratio of 1:4 (DNA:Lipofectamine), with the medium changed 6 h after cell culture inoculation.

## Transduction

Transduction of LVPs was obtained in the presence of 8 µg/mL polybrene. The medium was replaced after 16 h to remove unbound virus particles. At 24 h post-transduction, cells were cultured with fresh 10% DMEM containing 1 µg/mL puromycin (pLKO.1-shRNA LVPs) or 10 µg/mL blasticidin (pLV-Tat-IRES-Rev LVPs).

## Leptomycin B treatment

U2OS stable cell lines were transfected with pTat101 and pRev-GFP using Lipofectamine 2000 Reagent. After 24 h, the medium was refreshed and supplemented with leptomycin B (LMB) (20 nM, Abcam, ab120501) or an equal volume of ethanol as a negative control. Cells were placed in an incubator for 24 h and then collected by trypsinization. Half the volume of collected cells was fixed in 3.7% PFA and subjected to flow cytometry. Simultaneously, the other halves were lysed using RIPA buffer (50 mM Tris-Cl; pH 7.5, 1% NP-40, 0.05% SDS, and 150 mM NaCl) and subjected to western blotting.

## shRNA transduction

Depletion of protein expression was performed by transduction using previously published lentiviral vectors containing shRNA sequences for shLuc (control) or shMATR3 (24). Clones 4G10 and 6A12 were transduced using 20 µL of 40 times concentrated lentiviral vectors in the presence of polybrene (8 µg/mL) for 20 h. Next, the medium was changed to D10 containing puromycin at a concentration of 1 µg/mL for 3 days. After selection, cells were counted and plated for transfection using pTat-101 and pRev-GFP plasmids on the following day.

## siRNA knockdown

U2OS-dual clones 4G10 and 6A12 with stable expression of Tat-IRES-Rev transgene were subjected to SMARTpool siRNA transfection. The transfection was performed using Lipofectamine RNAiMAX Transfection Reagent (Invitrogen, 13778150) according to the manufacturer's protocol with modifications. Shortly, U2OS-dual clones were plated on 6-well plates till reaching 40% confluency, followed by two day-to-day siRNA transfections at 20 pmol of control siRNA (Dharmacon, D-001810-10-05) or siCRNKL1 (Dharmacon, L-019013-02-0005). The day after the second siRNA transfection, cells were subjected to further analysis.

## RESULTS

### Development and characterization of a double fluorescent reporter pHIV-dual

To establish a Rev-dependent system, we utilized the previously described HIV-1-derived vector pHIV_Intro (22). This vector contains the *cis*-acting sequences necessary for viral gene expression and subsequent replication steps, including the 5′ long terminal repeat (LTR), the Tat-responsive region TAR, the major splice donor (SD1), the packaging signal ψ, a portion of the gag gene encoding the p17 protein, the Rev-responsive element, the splice acceptor SA7 with its regulatory sequences (ESE and ESS3), and the 3′ LTR for 3′-end formation (Fig. 1A). Additionally, pHIV_Intro harbors the reporter gene positioned downstream of the splice acceptor SA7, encoding the cyan fluorescent protein fused with a peroxisome localization signal (ECFPskl). This vector is also tagged with 24 MS2 stem-loops in the intronic region, enabling RNA pull-down and visualization via the MS2-bacteriophage binding protein (MBP), as previously demonstrated (21).

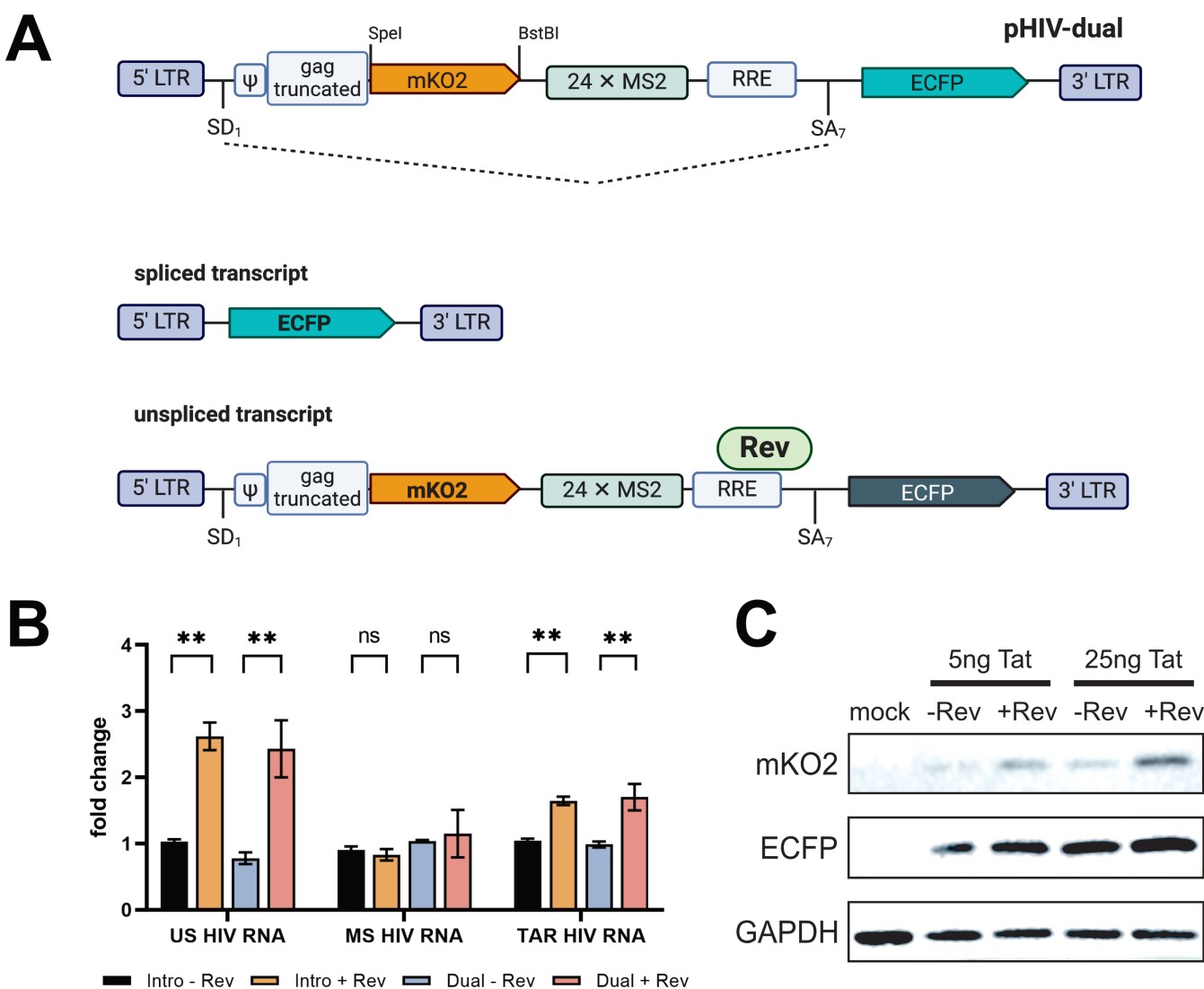

**FIG 1** Construction and validation of pHIV-dual vector. (A) The *mko2* gene was inserted into the intronic region of the pHIV_Intro construct at the SpeI and BstBI restriction sites, positioned just after the truncated gag gene. In the absence of Rev, the unspliced HIV-1 RNA undergoes splicing, resulting in a short, spliced transcript that contains the ecfp gene, leading to the expression of the ECFPskI protein. When Rev is present, it binds to the RRE sequence located within the intronic sequence, facilitating the export of the unspliced transcript from the nucleus and resulting in the expression of the mKO2 protein. (B) U2OS cells were transfected with the viral vectors and pTat-101, both in the absence and presence of the pRev-GFP vector. Twenty-four hours post-transfection, total RNA was isolated, and RT-qPCR was conducted to measure levels of unspliced, spliced, and TAR-containing total RNAs (primers are listed in Fig. S1). Values were normalized using GAPDH mRNA primers and are presented as fold changes relative to the samples without Rev, which were arbitrarily set to a value of 1. Results are displayed as mean values ± standard error of the mean. Statistical analysis was performed using the *t*-test with Welch correction from three biological replicates ($N = 3$), with two technical replicates each. Significance is indicated as follows: ns, not significant and **$P$-value < 0.05. (C) Immunoblotting was performed to detect mKO2 and ECFP levels in untransfected (mock) cells and cells transfected with the pHIV_dual vector, using two doses of the Tat-expressing plasmid, with or without the plasmid encoding Rev. GAPDH was used as a loading control.

Furthermore, we engineered the HIV-dual vector by inserting another reporter gene, mKO2, encoding the monomeric fluorescent protein Kusabira-Orange 2 (mKO2) into the intronic region immediately following the truncated gag gene (Fig. 1A). In the presence of Rev, which binds to the RRE sequence within the intronic region, the unspliced transcript is exported from the nucleus, resulting in the production of the mKO2 and ECFPskI fluorescent proteins (Fig. 1A). Conversely, in the absence of Rev, only the spliced transcript is exported, leading to the translation of the ECFPskI protein (Fig. 1A). Thus, mKO2 serves as a marker for Rev-dependent export, whereas ECFPskI indicates active

transcription, splicing, and translation. To confirm that the insertion of mKO2 did not disrupt the expression of unspliced HIV-1 RNA, we transfected U2OS cells with either pHIV-dual or the parental pHIV-Intro vector in the presence of pTat-101, with or without a Rev-expressing vector. We then performed RT-qPCR analysis to quantify (i) unspliced (Rev dependent), (ii) spliced (Rev independent), and (iii) total HIV-1 RNA levels. As shown in Fig. 1B, in the presence of Rev, unspliced transcript levels from HIV-dual were comparable to those from pHIV-Intro. As expected, spliced HIV-1 RNA levels were unaffected by Rev (Fig. 1B). Total transcript levels increased upon Rev, similar to the levels observed with the parental pHIV-Intro vector (Fig. 1B). Thus, the insertion of the *mko2* gene did not disrupt either unspliced or spliced HIV-1 RNA expressions from pHIV-dual. We also assessed mKO2 protein levels in the presence and absence of Rev using western blotting. As illustrated in Fig. 1C, although there was some leakage of mKO2 expression in the absence of Rev, mKO2 levels significantly increased upon Rev addition, while ECFP levels remained unchanged.

## Generation of clonal cell lines carrying a stably integrated inducible HIV-1-based double-fluorescent reporter vector

Next, we wished to generate a cell line with a stably integrated pHIV-dual reporter system. As illustrated in Fig. 2A, U2OS cells were transduced with HIV-dual lentiviral particles and subjected to FACS. During FACS, single cells negative for mKO2 and ECFP fluorescence (indicating either untransduced or transcriptionally silent cells) were sorted into 96-well plates. The resulting expanded clones were screened for integrated vector presence through a reactivation assay using a cocktail of PMA + ionomycin, TNFα, and transfection with a Tat-expressing vector to fully activate viral transcription. Fluorescence microscopy analysis for ECFPskl detection identified 23 clones carrying the inducible HIV-dual reporter vector (data not shown). The identified clones were further characterized for their Rev-dependent expression of the mKO2 reporter protein. As such, clones were transfected with pTat-101 in the presence or absence of pRev-GFP and subjected to flow cytometry analysis to measure the percentages (Fig. 2B) and mean fluorescent intensities (MFIs) (Fig. 2C) of mKO2-positive cells. These analyses revealed a heterogeneous profile of mKO2 expression across the clones, with seven clones showing Rev-independent phenotypes, and the remaining 16 displaying Rev-dependent phenotypes. From these 16 Rev-dependent clones, we arbitrarily selected five (2D5, 4F4, 4G10, 6A12, and 6B12) that exhibited ≥10% and ≥5,000 MFI of mKO2$^+$ cells upon Rev presence (Fig. 2B and C, respectively; selected clones are marked with "#"). Western blot analysis confirmed that these selected clones exhibited Rev-dependent expression of mKO2 (Fig. 2D). Notably, in some clones, the presence of Rev led to decreased ECFP protein expression, which could be attributed to the reported negative impact of Rev on splicing (27).

## Flow cytometry analyses to quantify Rev-dependent export efficiency in 4G10 and 6A12 clones

Next, we aimed to establish a robust strategy based on flow cytometry analysis to assess Rev-dependent export efficiency in our system. To this end, we transfected 4G10 and 6A12 clones with either pTat101 alone or co-transfected with pTat101 and pRev-GFP. As shown in Fig. 3, untransfected clones were negative for both fluorescent proteins indicating their transcriptionally silent status (Fig. 3A, left panels); however, they could be activated by transfection with pTat101 as shown by the appearance of ECFP-positive population (Fig. 3A, middle panels) and increases in the MFI for ECFP (Fig. 3B). Importantly, the presence of Rev led to the appearance of mKO2$^+$ECFP$^+$ double-positive population (Fig. 3A, right panels) and increases in the MFI for mKO2 (Fig. 3B). Of note, higher doses of Rev did not further impact the percentages of mKO2$^+$ECFP$^+$ double-positive cell populations (Fig. S1). To assess the fraction of cells actively responsive to Rev, we took advantage of leptomycin B, which is a well-described inhibitor of the CRM-1-dependent pathway utilized by Rev. To this end, 4G10 and 6A12 clones were

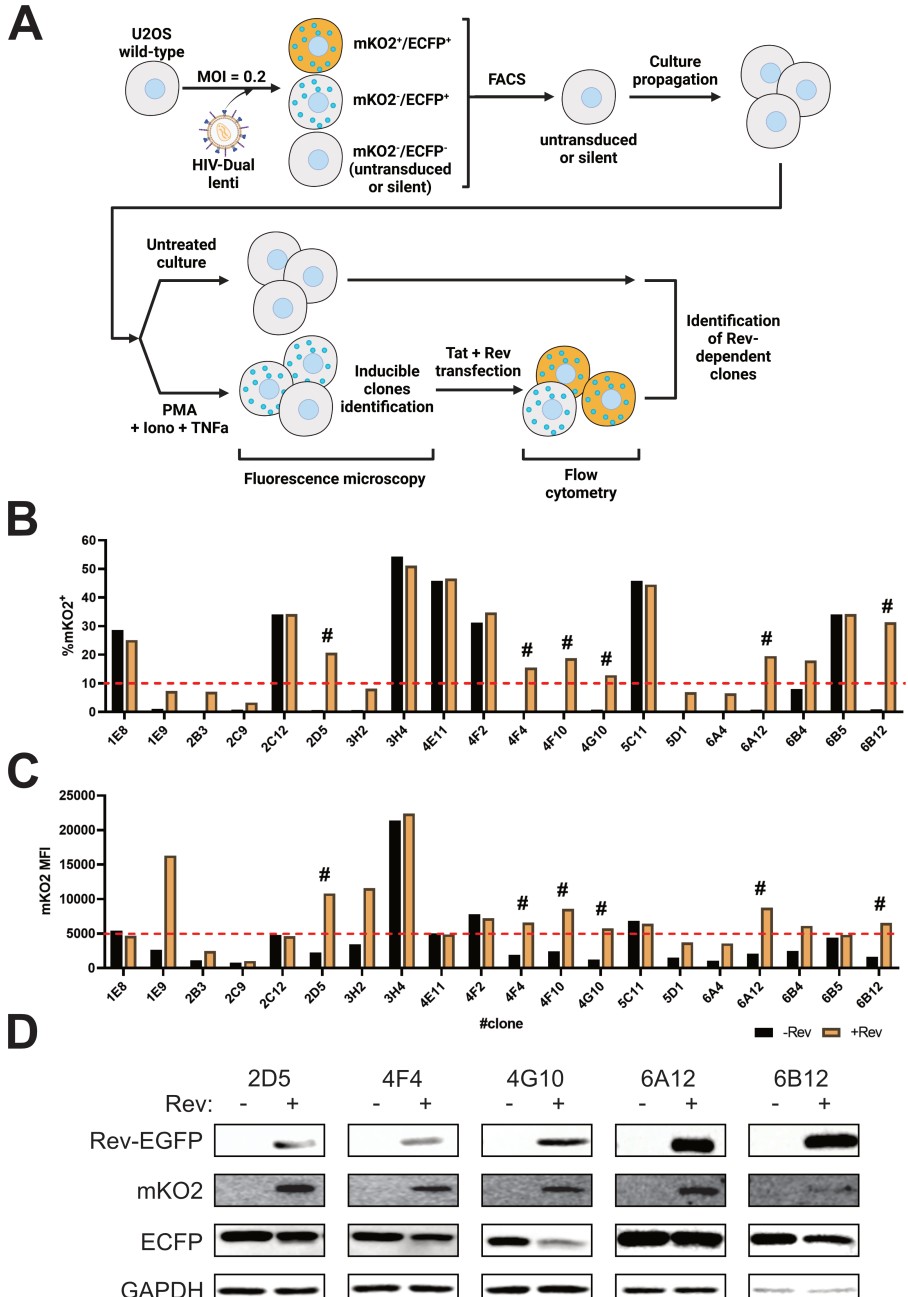

**FIG 2** Generation and validation of U2OS HIV-dual clone cell lines. (A) Schematic representation of screening protocol by fluorescent-activated cell sorting, fluorescent microscopy, and flow cytometry. See text for details. (B and C) Clonal cell lines isolated using the procedure described above were analyzed by flow cytometry to assess the (**B**) percentages and (C) mean fluorescent intensities of mKO2$^+$ cells transfected with pTat101 in the absence and presence of pRev-GFP. # indicates clones that were selected based on the following arbitrarily chosen criteria: ≥10% of mKO2$^+$ cells and ≥5,000 MFI of mKO2$^+$ cells. (D) Western blotting analysis of 2D5, 4F4, 4G10, 6A12, and 6B12 clonal cell lines to detect the levels of mKO2, Rev-GFP, and ECFP in pTat101-transfected cells in the absence and presence of pRev-GFP. GAPDH is the protein loading control.

transfected with pTat101 and pRev-GFP in the presence or absence of LMB and subjected to flow cytometry and western blot analysis. As shown in Fig. 4A, the presence of Rev led to the appearance of a double-positive population. The addition of LMB decreased the percentages of mKO2$^+$ECFP$^+$ cells in favor of increases in single-positive mKO2$^-$ECFP$^+$ cells. This shift toward single-positive mKO2$^-$ECFP$^+$ cells is a result of downregulation

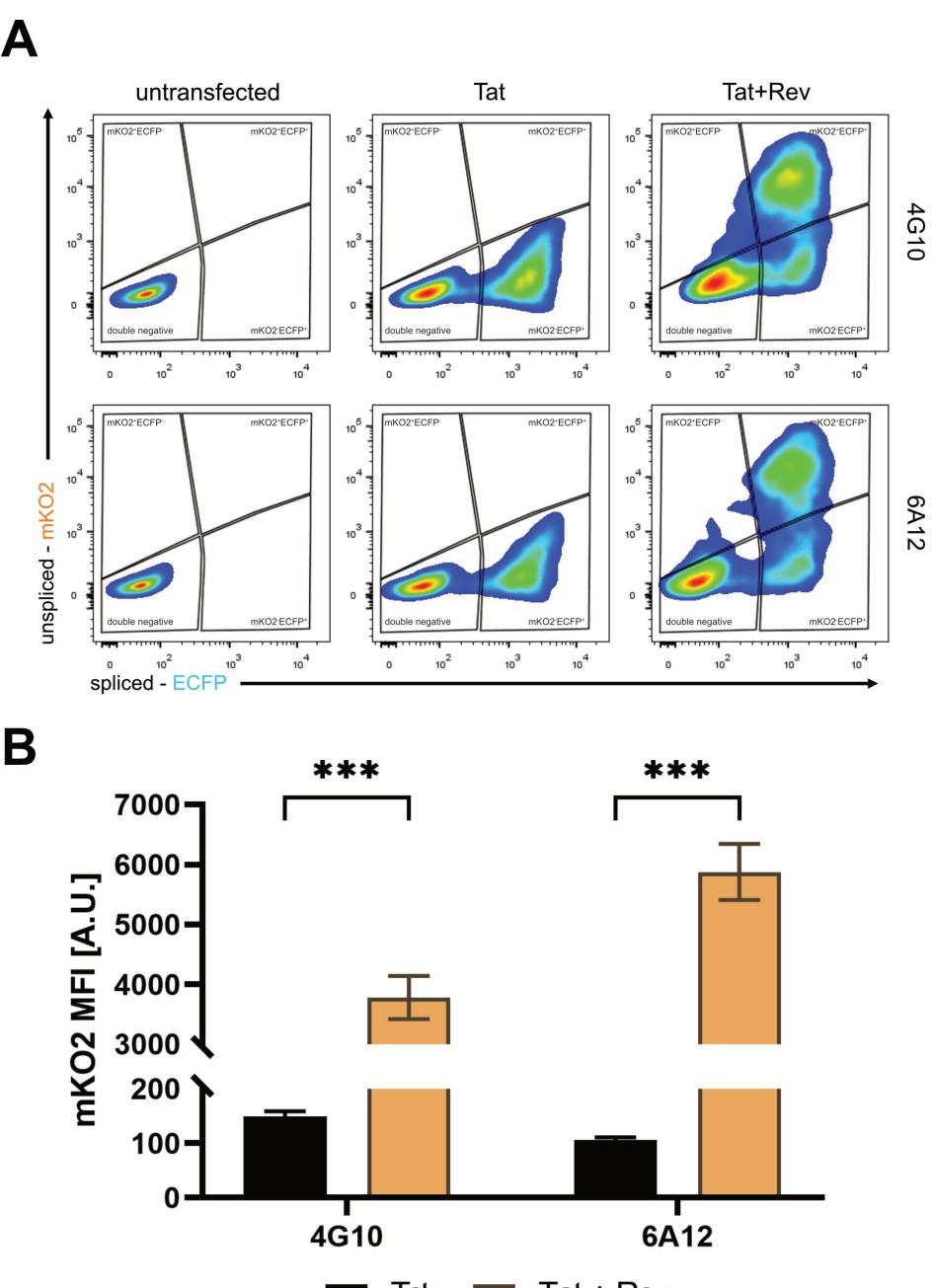

**FIG 3** Flow cytometry analysis of 4G10 and 6A12 clones. (A) Representative dot plots showing flow cytometry analysis on total cells of clones either untransfected (left panels), transfected with pTat101 alone (middle panels), or in the presence of Rev-GFP expressing vector (right panels). (B) Quantitative analysis of mKO2 median fluorescence intensity from ECFP⁺ cells is shown as mean values ± SD from two independent biological repetitions performed in two technical repetitions. Statistical analysis was performed using *t*-test with Welch correction, ***P-value < 0.001.

in mKO2 expression as confirmed by MFI analysis (Fig. S4A) and western blot (Fig. 4B). Next, to properly determine the observed shift that reflects the efficiency of LMB in the export inhibition, we set a threshold within the double-positive cells that was based on the MFI for mKO2 (Fig. S4A). As such, we distinguished two subpopulations of mKO2$^{high}$ECFP⁺ and mKO2$^{low}$ECFP⁺ cells, with the former being highly responsive to Rev (Fig. 4A). Next, evaluation of the percentages of mKO2$^{high}$ECFP⁺ cells that are highly responsive to Rev relative to the total number of ECFP+ cells provided an estimate of the

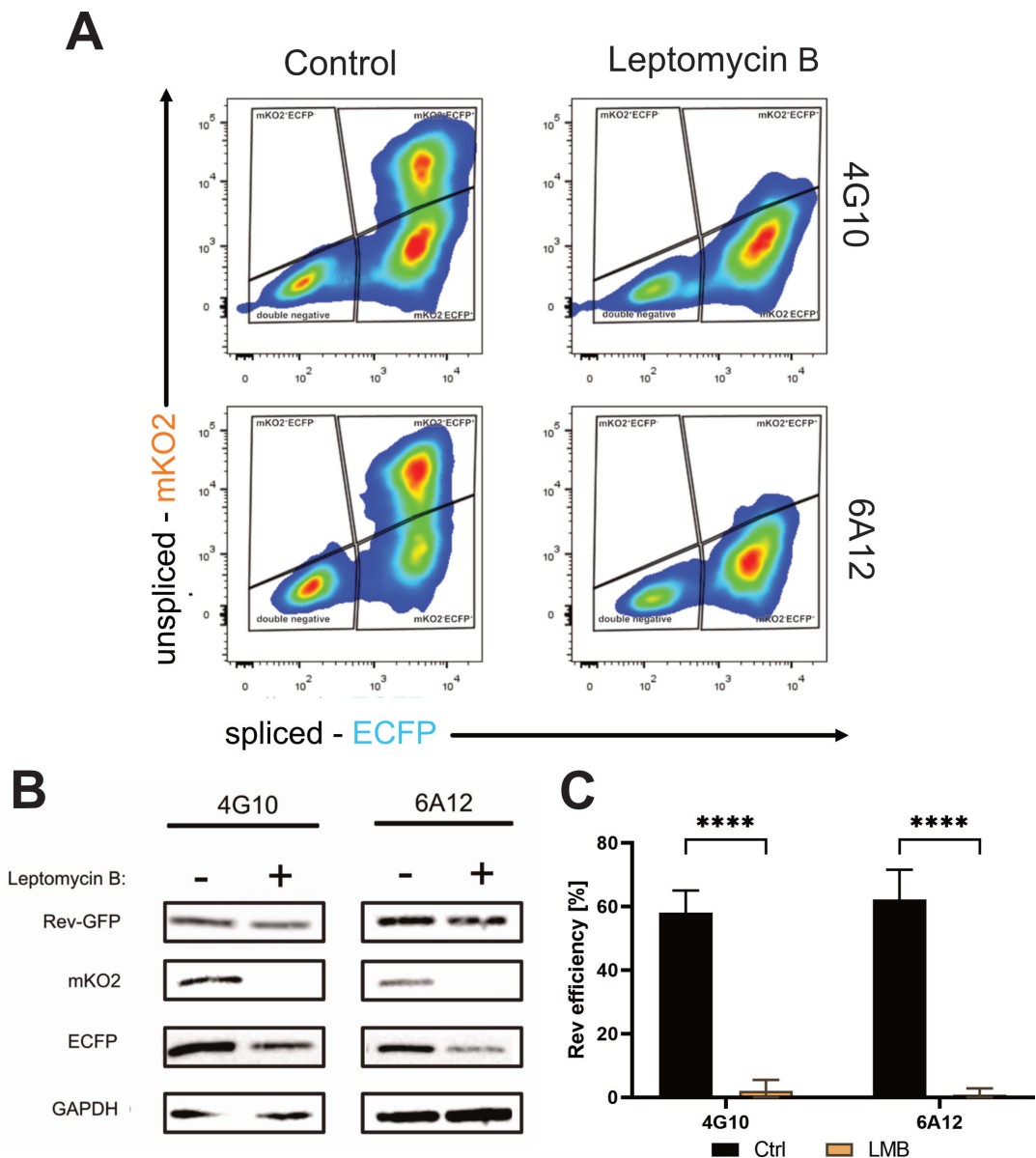

**FIG 4** Leptomycin B treatment of 4G10 and 6A12 clones results in decreased Rev efficiency. (A) Representative flow cytometry plots showing mKO2 vs ECFP fluorescence of mock-treated (left panels) or LMB-treated (right panels) U2OS_dual clones co-transfected with Tat and Rev-GFP plasmids. (B) Western blot analysis of the Rev-GFP, mKO2, and ECFP protein levels in mock-treated or LMB-treated cell lines. GAPDH is a loading control. (C) Evaluation of Rev efficiency (% mKO2$^+$high ECFP$^+$)/(% total ECFP$^+$) × 100. Data represent averages of five biological repetitions, and error bars indicate the standard error of the mean. Statistics are performed using $t$-test with Welch correction, ****$P$-value < 0.000001.

fraction of cells with active Rev-dependent export as reflected by the following equation: (% mKO2$^{high}$ECFP$^+$ cells)/(% total ECFP$^+$ cells) × 100. LMB was very potent in decreasing Rev efficiency in both clones as shown in Fig. 4C. Importantly, we also tested the effect of ABX-464, a known compound that inhibits Rev function (28). As shown in Fig. S3, ABX-464 treatment decreased the percentage of mKO2$^+$ cells population (Fig. S3C) and mKO2 protein levels (Fig. S3D), as addressed by flow cytometry and western blotting, respectively. However, due to the high autofluorescence of the compound in the ECFP spectrum, as shown by flow cytometry (Fig. S3B), quantification of the Rev efficiency was precluded.

Altogether, the dual-reporter HIV-1-based cellular system represents a useful tool to quantify Rev-dependent export and efficacy of compounds by means of flow cytometry.

## Depletion of cellular factors MATR3 and CRNKL1 modulates Rev-dependent export efficiency in dual reporter HIV-1-based cellular clones

To further validate our system, we aimed to modulate the expression of the Rev co-factor MATR3 (22, 23) in dual clones to assess its impact on Rev-mediated RNA export. MATR3 was depleted in the 4G10 and 6A12 clones with shRNA lentiviral vectors (24). After 3 days of puromycin selection, cells were transfected with pTat and pRev-GFP. Twenty-four hours post-transfection, cells were analyzed by flow cytometry and western blot to assess the impact of MATR3 depletion on Rev-dependent export. Western blot analysis confirmed that MATR3 depletion led to decreased mKO2 protein levels (Fig. 5A), corroborating previously published effects on Rev function (21–23). Depletion of the MATR3 resulted in a shift of the mKO2-positive population toward lower fluorescence intensity (Fig. 5B and C; Fig. S5A). Moreover, MATR3 levels were found to correlate with the percentages of mKO2-positive cells, whereas no correlation was observed with the percentages of ECFP-positive cells (Fig.S5B and C). Finally, we calculated the Rev efficiency upon MATR3 knockdown and observed a significant decrease in both analyzed clones (Fig. 5C).

Similarly, we aimed to modulate another recently identified cellular factor CRNKL1, which plays a role in the nuclear retention of Rev-responsive intron-containing viral RNAs (19). We took advantage of a previously validated siRNA pool from Xiao et al. (19) to deplete CRNKL1 in Tat and Rev co-transfected U2OS-dual 4G10 and 6A12 clones. However, due to impaired Rev-GFP transfection efficiency following siCRNKL1 treatments (data not shown), we introduced a stable simultaneous overexpression of the viral proteins Tat and Rev via a specially designed Tat-IRES-Rev lentiviral vector. As expected, CRKLN1 knockdown upregulated mKO2 protein levels as assessed by western blot in both clones (Fig. 6A). Next, we assessed the double-positive populations by flow cytometry in 4G10 and 6A12 clones. We showed that CRKLN1 knockdown resulted in a shift of the mKO2-positive population toward higher fluorescence intensity (Fig. 6B and C). Additionally, we calculated the Rev efficiency upon CRKNL1 knockdown and observed a significant increase in both analyzed clones (Fig. 6D) corroborating previously published results (19).

## DISCUSSION

Antiretroviral therapy cannot cure HIV infection because it is unable to eliminate latently infected cells. ART inhibits different steps of HIV-1 replication but does not target post-transcriptional HIV-1 RNA biogenesis (29). Moreover, experimental cure strategies aiming at latency elimination, such as "shock-and-kill," are ineffective in clinical trials (30–33). Latency-reversing agents of "shock-and-kill" aim to reactivate the latent virus for its subsequent elimination by viral cytopathic effects or immune cell recognition (33). However, LRAs target epigenetic and transcriptional blocks but overlook the poorly characterized post-transcriptional mechanisms, which may contribute to their ineffectiveness (34). Indeed, mechanisms acting at the transcriptional level have been preferably studied against post-transcriptional pathways (35, 36). Hence, more studies are needed to shed light on poorly characterized post-transcriptional mechanisms to identify new anti- and pro-viral drugs for viral elimination.

In this study, we developed a novel dual-fluorescent HIV-1-based reporter system to elucidate the mechanisms underpinning Rev-dependent export of HIV-1 RNA. Rev is a critical factor required for productive HIV-1 infection. It is crucial for the nuclear export of US, full-length genomic HIV-1 RNA, which is packed into newly produced virions and translated into HIV-1 structural proteins, as well as SS HIV-1 RNA that is translated into HIV-1 accessory proteins. Our dual reporter system, by the expression of mKO2 and ECFP fluorescent proteins, allows for a distinction between Rev-dependent and Rev-independent RNA processing events. Currently, there are no approved LRAs and no anti-HIV drugs that would target post-transcriptional HIV-1 RNA processing. Several classes of RRE-Rev inhibitors and Rev binders were identified but had limited antiretroviral activity (37, 38).

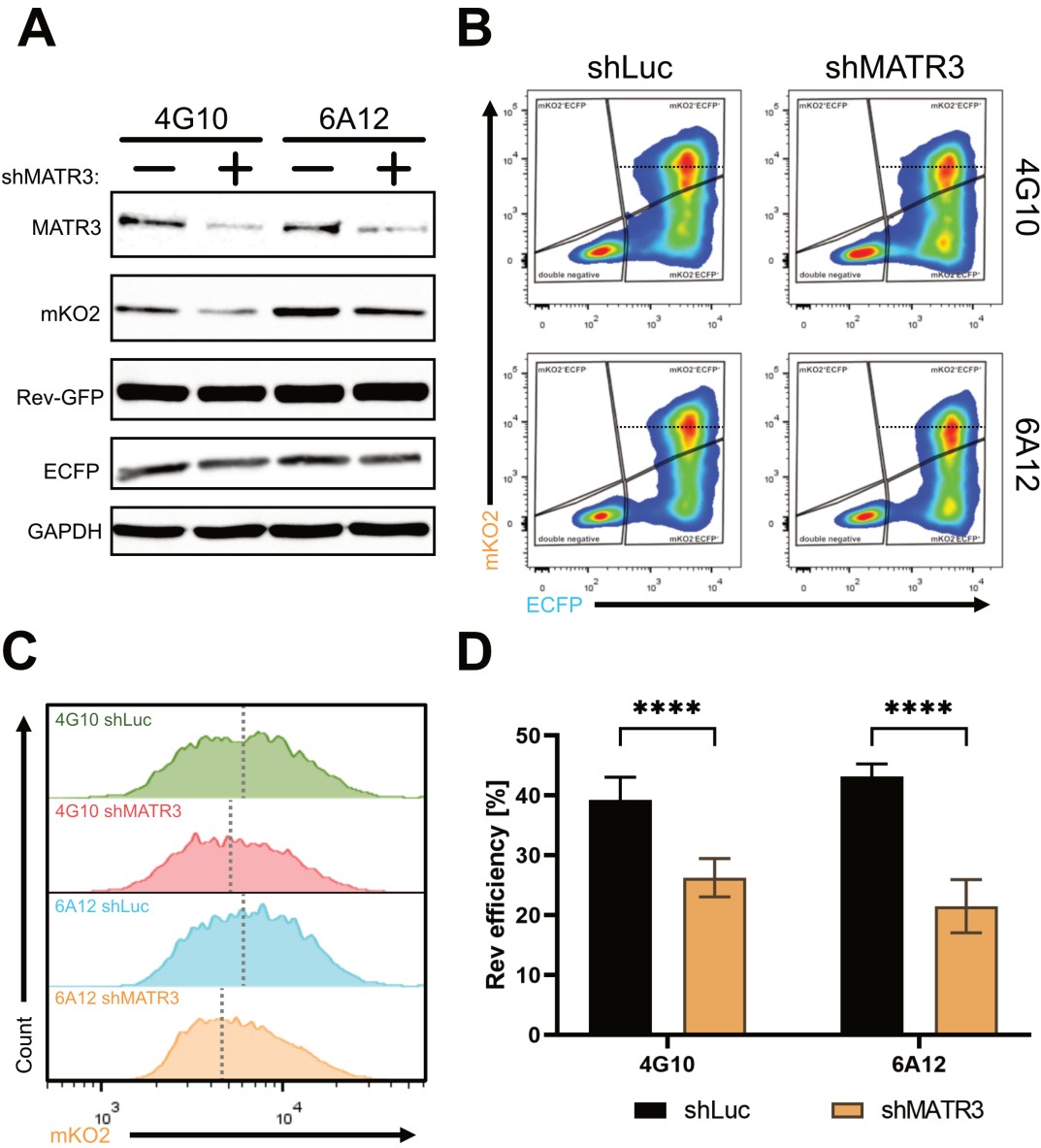

**FIG 5** Depletion of MATR3 decreases Rev activity. Clones 4G10 and 6A12 were transduced with lentiviral vectors containing shRNA against luciferase (shLuc, control) or MATR3 (shMATR3). After 3 days of puromycin (1 ng/µL) selection, cells were co-transfected with pTat-101 and pRev-GFP plasmids. Twenty-four hours after transfection, cells were collected for western blotting (A) and flow cytometry (B–D) analysis. (A) Western blot analysis to detect MATR3, mKO2, Rev-GFP, and ECFP protein levels in shLuc- and shMATR3-transduced cells. GAPDH is a loading control. (B) Representative flow cytometry plots showing mKO2 vs ECFP fluorescence of control shLuc-transduced (left panels) or shMATR3-transduced (right panels) U2OS_dual clones. The black dotted line indicates the median mKO2 fluorescence from shLuc cells within the double-positive population, distinguishing two subpopulations: mKO2$^{high}$ and mKO2$^{low}$. (C) Comparison of mKO2 fluorescence histograms from double-positive population shown in panel B. The gray dotted line indicates the median fluorescence intensity of mKO2. (D) Evaluation of Rev efficiency (% mKO2$^+$high ECFP$^+$)/(%total ECFP$^+$) × 100. Data represent averages of three biological repetitions ($N = 3$), and error bars indicate the standard error of the mean. Statistics are performed using $t$-test with Welch correction, ****$P$-value < 0.0001.

Most of these compounds, however, did not block RRE-Rev binding *in vitro* (37, 39) and thus may act with different mechanisms. In this context, ABX464 (also called ABIVAX or obefazimod), a first-in-class anti-HIV compound inhibiting Rev function by binding to the cap-binding complex, altered HIV-1 RNA biogenesis *in vitro* and *ex vivo* (28). Moreover, in a recent phase 2a clinical trial ABX464-005 (NCT02990325), ABX464 showed a reduction in the total HIV reservoir and HIV transcription initiation in CD4$^+$ T cells from ART-treated HIV$^+$ individuals, suggesting that ABX464 is a promising new antiviral

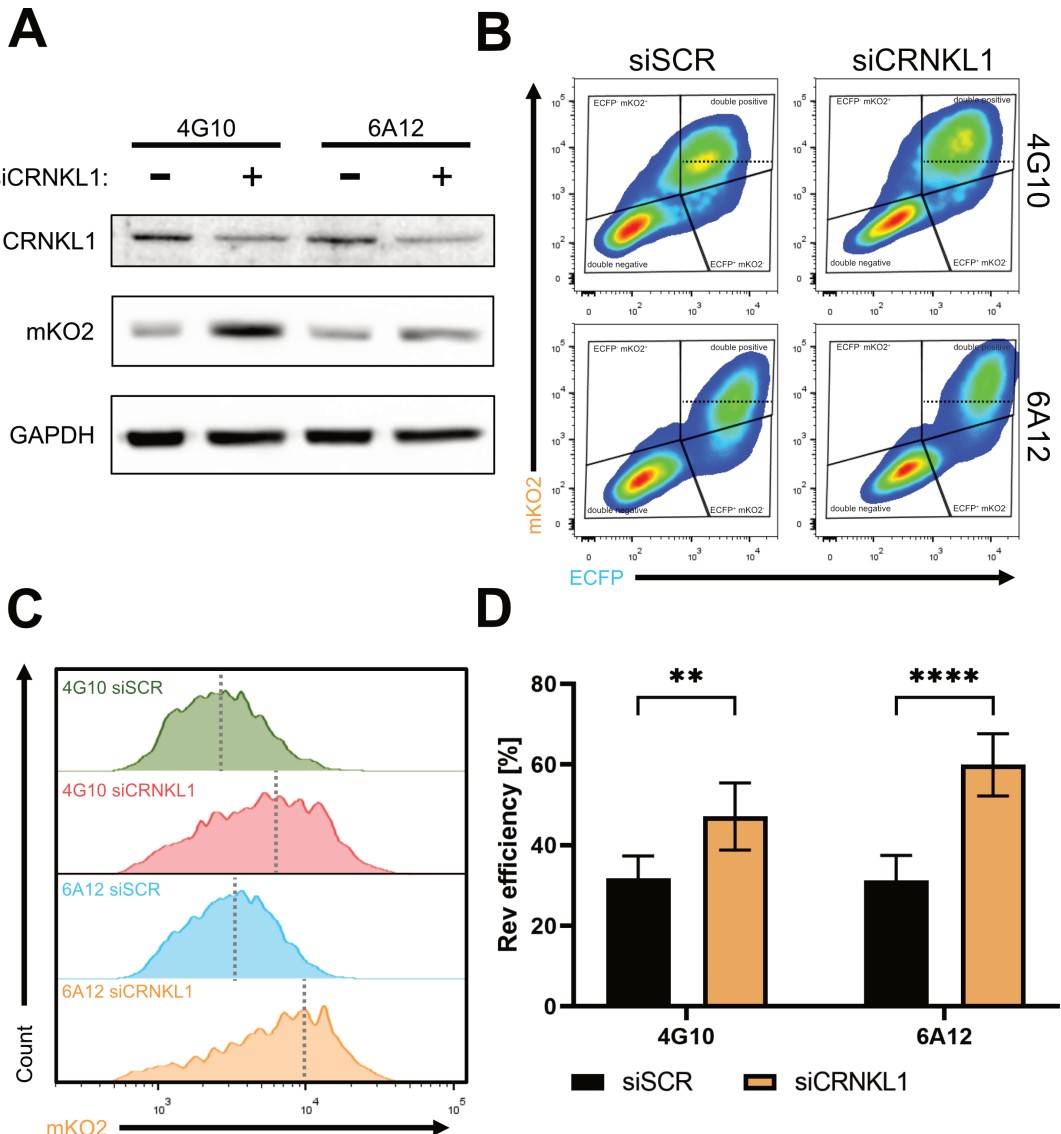

**FIG 6** Depletion of CRNKL1 enhances Rev activity. Clones 4G10 and 6A12 were transduced with a lentiviral vector encoding Tat-IRES-Rev transgene. After 3 days of blasticidin (10 µg/µL) selection, cells were transfected with siSCR (scramble control siRNA) or siCRNKL1. At 24 h after siRNA transfections, cells were collected for (A–C) flow cytometry and (D) western blotting. (A) Western blot analysis to detect CRNKL1 and mKO2 protein levels in siSCR- and siCRNKL1-transfected cells. GAPDH is a loading control. (B) Representative flow cytometry plots showing mKO2 vs ECFP fluorescence of control (left panels) or CRNKL1-depleted (right panels) U2OS_dual clones. The black dotted line indicates the median mKO2 fluorescence from siSCR-treated cells within the double-positive population, distinguishing two subpopulations: mKO2$^{high}$ and mKO2$^{low}$. (C) Comparison of mKO2 fluorescence histograms from double-positive population shown in panel B. The gray dotted line indicates the median fluorescence intensity of mKO2. (D) Evaluation of Rev efficiency (% mKO2$^+$high ECFP$^+$)/(%total ECFP$^+$) $\times$ 100. Data represent averages of three biological repetitions, and error bars indicate the standard error of the mean. Statistics are performed using $t$-test with Welch correction, **$P$-value < 0.01 and ****$P$-value < 0.0001.

drug (40). However, its specific mechanism of action is unclear as it was also shown to induce the splicing of lncRNA 0599-205, resulting in the generation of anti-inflammatory microRNA miR-122 (41, 42). Therefore, ABX464 is rather not a specific Rev inhibitor. Thus, more effort is needed to search for new promising specific Rev modulators. Our platform can be conveniently used for high throughput screenings of compounds modulating Rev activity by assessing the expression of the ECFP and mKO2 markers.

Although Rev and its associated factors have been extensively studied, several aspects of its mechanism remain incompletely understood. Identifying novel factors that interact with Rev and refining tools to investigate its role are essential for dissecting

less-characterized post-transcriptional stages of HIV-1 RNA metabolism in latency and reactivation. We showed that the depletion of MATR3, a known Rev cofactor, significantly decreased mKO2 levels and hampered Rev function in our system. The reduction in mKO2 fluorescence intensity and protein levels upon MATR3 knockdown confirmed its role in the Rev-mediated export pathway. On the other hand, the knockdown of CRKNL1, a known factor retaining unspliced HIV-1 RNA in the nucleus, increased Rev activity in our system. These findings align with previous reports (19, 22, 23) and further validate the reliability of our dual reporter system in studying host factors involved in HIV-1 RNA processing. It is likely that the depletion of host factors will result in less robust phenotypes unlike the more pronounced effects seen with compound inhibition as we observed in the case of MATR3 depletion vs LMB treatment. This could be explained by the activation of cellular compensatory mechanisms that could at least partially reverse the phenotype. The modest phenotype observed with MATR3 and CRKLN1 knockdowns suggests that these factors are part of a broader network of host factors involved in the nuclear retention and export of unspliced HIV-1 RNAs. Additional cellular components, potentially interacting with MATR3 or acting independently, may also play critical roles in these processes.

Limitations of the study should also be acknowledged. First, we used U2OS cells that are not a physiological cellular target for HIV-1. However, adherent U2OS cells represent an attractive model system due to their robust growth (43, 44), which should be beneficial for future high-throughput screening studies. Second, we utilized a minimal viral vector rather than a full-length virus. While the absence of significant portions of the HIV-1 genome—including splice sites SD2-SD4, SA1-SA5, and splicing regulatory sequences—limits our model's ability to capture the complexity of HIV-1 gene regulation, its potential utility in studying specific splicing events warrants further investigation. However, this approach makes the system more robust and specifically tailored for studying the Rev-dependent pathway, without interference from other events associated with a full-length virus. This system should be considered as a first-line screening tool, and validation of the results is required by using full-length viruses in relevant cellular systems of CD4$^+$ cells. Autofluorescence of compounds should also be considered as it could mask their specific effects, as we observed in the case of ABX464. ABX464 belongs to quinolines that contain an aromatic ring structure that can absorb light and emit fluorescence in the visible range when excited by ultraviolet or visible light. In fact, this property makes quinoline and its derivatives useful in fluorescence microscopy and other analytical techniques as fluorescent chemosensors (45, 46).

Several HIV-1 replication models exploiting two fluorescent proteins have been already established in the field, including HIV-GKO (Green–Kushibara Orange)(47), DFV-B (dual fluorescent virus-B) (48), RGH (Red-Green-HIV-1) (49), 89mASG (89.6/DNE/mApple/ SFG) and R7GEmC (R7/E-/GFP/EF1a-mCherry) (50), and Duo-Fluo (51). These models, based on nearly full-length HIV-1, are used to mark latently infected cells, as one fluorescent protein is under LTR regulation and the second under an independent constitutive promoter. Another recent model called HIV-dual-GT (HIV-dual-GagBFP-Tat) (19) has been introduced in which two fluorescent proteins come either from US (Gag-BFP) or MS (mCherry in place of Nef) viral transcripts. This model contains multiple point mutations localized in a majority of the viral genes, leaving only *gag* and *tat* intact, but all known *cis*-acting sequences potentially interacting with cellular components are maintained (19). HIV-dual-GT is similar to our simplified HIV-dual as it addresses Rev-dependent and Rev-independent expressions.

In conclusion, the HIV-dual fluorescent reporter system we developed here represents a convenient and versatile tool for dissecting the molecular intricacies of Rev-dependent RNA export by identifying new cellular factors involved in this process. Additionally, this system holds great potential for high-throughput drug screening, providing a valuable platform for discovering novel therapeutic agents aimed at disrupting the post-transcriptional stages of the HIV-1 life cycle.

## ACKNOWLEDGMENTS

A.K.P. and J.W. acknowledge funding from the National Science Centre, Poland (OPUS Grant UMO-2022/45/B/NZ3/03890). We thank Neli Kachamakova-Trojanowska and Dawid Skoczek (Malopolska Centre of Biotechnology, Jagiellonian University) for excellent technical assistance with FACS.

A.K.P. envisioned the project and supervised the work. J.W. performed most of the experiments. H.A. helped with flow cytometry data analysis. A.O. helped with western blotting analysis. A.D. helped with cloning experiment. A.K.P. and J.W. wrote the paper and all authors discussed and edited the paper. K.P. revised the manuscript.

## AUTHOR AFFILIATIONS

[1]Laboratory of Molecular Virology, Malopolska Centre of Biotechnology, Jagiellonian University, Krakow, Poland
[2]Doctoral School of Exact and Natural Sciences, Jagiellonian University, Krakow, Poland
[3]Laboratory of Virology–Virogenetics, Malopolska Centre of Biotechnology, Jagiellonian University, Krakow, Poland

## AUTHOR ORCIDs

Jakub Wadas  http://orcid.org/0000-0003-1675-6286
Krzysztof Pyrc  http://orcid.org/0000-0002-3867-7688
Anna Kula-Pacurar  http://orcid.org/0000-0001-8404-2176

## FUNDING

| Funder | Grant(s) | Author(s) |
| --- | --- | --- |
| Narodowe Centrum Nauki (NCN) | UMO-2022/45/B/NZ3/03890 | Anna Kula-Pacurar |
| Narodowe Centrum Nauki (NCN) | UMO-2022/45/B/NZ3/03890 | Jakub Wadas |

## AUTHOR CONTRIBUTIONS

Jakub Wadas, Data curation, Formal analysis, Methodology, Project administration, Validation, Writing – original draft | Haider Ali, Data curation, Methodology | Aleksandra Osiecka, Investigation, Methodology | Agnieszka Dorman, Methodology, Writing – original draft | Krzysztof Pyrc, Writing – review and editing | Anna Kula-Pacurar, Conceptualization, Data curation, Funding acquisition, Investigation, Project administration, Resources, Supervision, Writing – original draft

## ADDITIONAL FILES

The following material is available online.

### Supplemental Material

**Supplemental table and figures (Spectrum01903-24-S0001.docx).** Table S1; Fig. S1 to S5.

### Open Peer Review

**PEER REVIEW HISTORY (review-history.pdf).** An accounting of the reviewer comments and feedback.

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
