## [Reviewer comments · Microbiology Spectrum]

Microbiology Spectrum

Development and characterization of a double-fluorescent HIV-1 reporter cellular model to tackle the Rev-dependent export pathway.

Anna Kula-Pacurar, Jakub Wadas, Haider Ali, Agnieszka Suder, Krzysztof Pyrc, and Aleksandra Osiecka

Corresponding Author(s): Anna Kula-Pacurar, Uniwersytet Jagiellonski w Krakowie

Review Timeline:

Submission Date:	July 30, 2024
Editorial Decision:	September 5, 2024
Revision Received:	December 20, 2024
Accepted:	January 9, 2025

Editor: Takamasa Ueno

Reviewer(s): Disclosure of reviewer identity is with reference to reviewer comments included in decision letter(s). The following individuals involved in review of your submission have agreed to reveal their identity: Sebla Bulent Kutluay (Reviewer #1)

Transaction Report:

DOI: <https://doi.org/10.1128/spectrum.01903-24>

Re: Spectrum01903-24 (Development and characterization of a double-fluorescent HIV-1 reporter cellular model to tackle the Rev-dependent export pathway.)

Dear Prof. Anna Kula-Pacurar:

Thank you for the privilege of reviewing your work. Below you will find my comments, instructions from the Spectrum editorial office, and the reviewer comments.

Both of our reviewers with expertise to the scope of the field essentially liked this work. Please address the issues raised by them.

Revision Guidelines

Sincerely,
Takamasa Ueno
Editor
Microbiology Spectrum

Reviewer #1 (Comments for the Author):

In this article Wadas et al. develop a dual reporter HIV-1 with mKO2 and ECFP that mark unspliced (i.e. Rev/RRE-dependent) and multiply spliced (i.e. Rev/RRE-independent) mRNA export. In addition to generating this reporter plasmid, the authors

generate U2OS single cell clones with stably integrated and transcriptionally silent versions of this reporter HIV-1 that can be reactivated by the viral transcriptional activator Tat as well as other latency reversing agents. Upon reactivations the reporter cells turn ECFP on but mKO2 positivity is fully dependent on the presence of Rev, the viral RNA export protein. Overall, this is a very well-written manuscript that generates a useful tool for the field but can be improved with the following revisions:

Minor:

- 1- Page 3: PMBCPBMC
- 2- Please include more details on the HIV-1 reporter construction in the Methods section including primers used.
- 3- Supplementary Fig 1: y-axis says % protein but I assume this is %cells.
- 4- Page 12: "As shown in Supplementary Figure 3"  Figure refers to Suppl. Fig. 4.
- 5- Page 12: "western" should not be capitalized.
- 6- Discussion has very long paragraphs, please break it down to coherent paragraphs.

Major:

- 1- Similar constructs have been developed by others and an example is PMC7845644. Perhaps the authors should elaborate on why their approach compares to some of the other published studies.
- 2-The effect of MATR3 knockdown seems to be very modest. An obvious interpretation, which I believe that the authors are interested in as well based on the proposed use of this assay in CRISPR screens, is that there are other factors involved in nuclear retention of unspliced HIV-1 RNAs and/or export. I suggest incorporating this discussion in both Results and Discussion.
- 3- Can the authors validate CRNKL1 knockdown/knockout in their assays? I think it would be useful for the field to have some sort of validation of the published work irrespective of the results.
- 4- Regarding MATR3 knockdown data, would it be possible to assess mKO2 vs. MATR3 in FACS to evaluate whether MATR3 low cells also express less mKO2?

Reviewer #2 (Comments for the Author):

The manuscript by Wadas et al. report the development of a cell-based assay system for the investigation of HIV-1 Rev function. The system generated consists of two fluorescent reporter proteins; mKO2 from Rev-dependent RNAs, and ECFP from Rev-independent RNAs. Reporters were introduced into a previously described HIV-1 derived vector (pHIV-Intro) in which a significant fraction of the HIV-1 genome (between Gag and the CA7 splice site) has been removed. In transient transfection assays, co-transfection of Rev increases HIV-1 US RNA abundance ~3 fold with minimal effect on the levels of the other viral RNAs examined (TAR, MS HIV RNA). Subsequent generation of stable cell lines containing this vector resulted in a mixed effect (Fig. 2B,C) in terms of percentage of cells displaying Rev-dependent fluorescence and the change in mean fluorescence-intensity; some of the cell lines showing a limited response to Rev while others have a strong response. The basis for such striking differences among the clones is not investigated but subsequent analyses with the clones conferring the desired phenotype did determine that mKO2 expression is Rev dependent while, for a subset of the clones, ECFP levels decreased with Rev addition. In subsequent flow cytometry studies, the authors demonstrate that, for two of the clones evaluated, addition of Tat significantly increases ECFP levels while addition of Tat and Rev results in both ECFP and mKO2 expression. Expression of mKO2 but not ECFP is reduced upon leptomycin B addition (an inhibitor of CRM1 function). In additional assays, the authors show that shRNA depletion of MATR3, a known Rev cofactor, also results in ~2 fold decrease in mKO2 levels.

Overall, the authors have generated an interesting reporter system to explore factors that could affect Rev function. Other groups have generated simpler systems in which fluorescent protein is expressed from either Gag or Env-encoding RNAs but this construct is the first to my knowledge that uses a dual reporter assay to look at Rev-dependent and independent expression from the same vector. One weakness of the system reported here is the loss of significant portions of the HIV-1 genome (splice sites SD2-SD4, SA1-SA5 and splicing regulatory sequences) that contribute to the complexity of HIV-1 gene regulation. Recent work by the Yukl group have highlighted the importance of splicing regulation to the HIV-1 latency phenotype. Although the authors demonstrate the responsiveness of their system to leptomycin B addition, no testing of HIV-1 splicing modulators (digoxin, sudemycin) is provided to examine the limitations of the system. The effect of depleting MATR3 using shRNA is consistent with previous reports but the magnitude of the effect raises questions regarding the utility of the system for more complex shRNA/CRISPR screens. It would have been interesting to demonstrate that the authors could enrich for shRNA sequences for MATR3 in a more complex library (i.e. 1 or 10% of the input library) as a demonstration of the suitability for such screens.

Major point

- 1) Contrary to the opening statement of the authors, I would consider HIV-1 Rev a well studied and well understood system. Multiple publications have explored the interaction between Rev and the RRE target RNA sequence, the cellular factors mediating the export of HIV-1 RNAs to the cytoplasm, and the composition of the exported RNA. The authors should acknowledge this point and appear to do in subsequent paragraphs.
- 2) Data in Fig. 3A and B appear inconsistent regarding the changes in mKO2 expression for clone 6A12. Flow shows a marked change in frequency of cells that are mKO2+ upon Rev addition in Fig. 3A but the MFI effect is only ~2 fold in Fig. 3B.
- 3) Blots in Fig. 5B regarding the changes in mKO2 expression do not appear to be consistent with the quantitation shown in Fig.

5C.

4) Supplementary Fig. 4. Western blots shown in Fig. 4B do not appear consistent with the quantitation in Fig. 4A. No double positive signal is shown in Fig. 4C with ABX464 addition but western blots show significant mKO2 expression in 4B.

Minor point

1) The opening paragraph of the introduction, at a minimum, should provide references to the primary papers that have detailed the mechanism by which Rev regulates HIV-1 gene expression in support of the statements made.

20th of December, Kraków

Dear Editor,

We warmly thank you for the reviewer reports for our manuscript "**Development and characterization of a double-fluorescent HIV-1 reporter cellular model to tackle the Rev-dependent export pathway**". We sincerely appreciate the valuable feedback and the opportunity to revise our manuscript. You will find enclosed a revised version of our manuscript and our point-by-point responses to the reviewers' criticisms and suggestions.

We have comprehensively addressed the comments from the two reviewers and fully acknowledged the concerns raised, especially regarding the validation of our system by depleting additional cellular factor, CRKNL1, that play a role in the retention of unspliced Rev-responsive viral transcripts (new Fig. 6). Moreover, as requested by the reviewer, we correlated the MATR3 levels with mKO2 expression by immunostaining using flow cytometry (new Supplementary fig. 6). We strongly believe that our manuscript has greatly improved as result.

We hope that you will find our manuscript suitable for *Microbiology Spectrum* and look forward to hearing from you.

Sincerely,

Anna Kula-Pacurar, PhD

Laboratory of Molecular Virology, Group Leader

Jagiellonian University (JU)

Małopolska Centre of Biotechnology (MCB)

Gronostajowa 7a

30-387 Kraków

Poland

Tel: 0048-579168571

e-mail: anna.kula-pacurar@uj.edu.pl

ANSWERS TO THE REVIEWERS

Following is a point-by-point response to the reviewers' comments. Reviewer's comments are shown *in italic*. Our responses are in **blue**.

Reviewer #1 (Comments for the Author):

In this article Wadas et al. develop a dual reporter HIV-1 with mKO2 and ECFP that mark unspliced (i.e. Rev/RRE-dependent) and multiply spliced (i.e. Rev/RRE-independent) mRNA export. In addition to generating this reporter plasmid, the authors generate U2OS single cell clones with stably integrated and transcriptionally silent versions of this reporter HIV-1 that can be reactivated by the viral transcriptional activator Tat as well as other latency reversing agents. Upon reactivations the reporter cells turn ECFP on but mKO2 positivity is fully dependent on the presence of Rev, the viral RNA export protein. Overall, this is a very well-written manuscript that generates a useful tool for the field but can be improved with the following revisions:

Minor:

1- Page 3: PMBCPBMC

3- Supplementary Fig 1: y-axis says % protein but I assume this is %cells.

4- Page 12: "As shown in Supplementary Figure 3"  Figure refers to Suppl. Fig. 4.

5- Page 12: "western" should not be capitalized.

6- Discussion has very long paragraphs, please break it down to coherent paragraphs.

We greatly appreciate the editorial notes and have addressed them accordingly in the revised manuscript. We also made changes in the discussion as suggested.

2- Please include more details on the HIV-1 reporter construction in the Methods section including primers used.

Thank you for this comment. As requested, we included more details in the M&M section (lines 140-148).

Major:

1- Similar constructs have been developed by others and an example is PMC7845644. Perhaps the authors should elaborate on why their approach compares to some of the other published studies.

We thank the reviewer for this insightful comment. As suggested, we have included a description of similar models incorporating dual fluorescent proteins (lanes 422-427). Additionally, we have acknowledged the HIV-dual-GT model developed by the Uberla laboratory (PMID: 33468685), which features fluorescent proteins derived from unspliced (Gag-BFP) and multiply spliced (mCherry replacing Nef) transcripts (lanes 427-433).

2-The effect of MATR3 knockdown seems to be very modest. An obvious interpretation, which I believe that the authors are interested in as well based on the proposed use of this assay in CRISPR screens, is that there are other factors involved in nuclear retention of unspliced HIV-1 RNAs and/or export. I suggest incorporating this discussion in both Results and Discussion.

We thank the reviewer for his/her comment. We agree that the modest effect of MATR3 knockdown likely reflects the involvement of additional host factors in the nuclear retention and/or export of unspliced HIV-1 RNAs as also recently highlighted by us (<https://doi.org/10.1016/j.jve.2024.100450>). We incorporated this discussion into the Results (lanes 82-83) and Discussion sections (lanes 400-404).

3- Can the authors validate CRNKL1 knockdown/knockout in their assays? I think it would be useful for the field to have some sort of validation of the published work irrespective of the results.

We thank the reviewer for his/her valuable comments. As suggested by the reviewer, we validated our system by depleting CRNKL1, and we confirmed that it upregulated Rev activity (new Figure 6) which is in line with previously published findings from Uberla laboratory.

4- Regarding MATR3 knockdown data, would it be possible to assess mKO2 vs. MATR3 in FACS to evaluate whether MATR3 low cells also express less mKO2?

Thank you for the suggestion. We immunodetected MATR3 with anti-MATR3 antibodies and subjected the cells to flow cytometry and correlated its levels with frequency of mKO2+ and ECFP+ cells (new Supplementary Fig. 5B-C). As expected, MATR3 levels were found to correlate with the percentages of mKO2+ cells, whereas no correlation was observed with the percentages of ECFP+ cells.

Reviewer #2 (Comments for the Author):

The manuscript by Wadas et al. report the development of a cell-based assay system for the investigation of HIV-1 Rev function. The system generated consists of two fluorescent reporter proteins; mKO2 from Rev-dependent RNAs, and ECFP from Rev-independent RNAs. Reporters were introduced into a previously described HIV-1 derived vector (pHIV-Intro) in which a significant fraction of the HIV-1 genome (between Gag and the CA7 splice site) has been removed. In transient transfection assays, co-transfection of Rev increases HIV-1 US RNA abundance ~3 fold with minimal effect on the levels of the other viral RNAs examined (TAR, MS HIV RNA). Subsequent generation of stable cell lines containing this vector resulted in a mixed effect (Fig. 2B,C) in terms of percentage of cells displaying Rev-dependent fluorescence and the change in mean fluorescence-intensity; some of the cell lines showing a limited response to Rev while others have a strong response. The basis for such striking differences among the clones is not investigated but subsequent analyses with the clones conferring the desired phenotype did determine that mKO2 expression is Rev dependent while, for a subset of the clones, ECFP levels decreased with Rev addition. In subsequent flow cytometry studies, the authors demonstrate that, for two of the clones evaluated, addition of Tat significantly increases ECFP levels while addition of Tat and Rev results in both ECFP and mKO2 expression. Expression of mKO2 but not ECFP is reduced upon leptomycin B addition (an inhibitor of CRM1 function). In additional assays, the authors show that shRNA depletion of MATR3, a known Rev cofactor, also results in ~2 fold decrease in mKO2 levels.

Overall, the authors have generated an interesting reporter system to explore factors that could affect Rev function. Other groups have generated simpler systems in which fluorescent protein is expressed from either Gag or Env-encoding RNAs but this construct is the first to my knowledge that uses a dual reporter assay to look at Rev-dependent and independent expression from the same vector. One weakness of the system reported here is the loss of significant portions of the HIV-1

genome (splice sites SD2-SD4, SA1-SA5 and splicing regulatory sequences) that contribute to the complexity of HIV-1 gene regulation. Recent work by the Yukl group have highlighted the importance of splicing regulation to the HIV-1 latency phenotype. Although the authors demonstrate the responsiveness of their system to leptomycin B addition, no testing of HIV-1 splicing modulators (digoxin, sudemycin) is provided to examine the limitations of the system.

Thank you for raising the question regarding the utility of our system for studying splicing. As requested by the reviewer, we evaluated the effect of the pre-mRNA splicing inhibitor Isoginkgetin (IGG) in our model and compared it to the J-Lat latency model, which contains a full-length virus. IGG treatment produced a similar phenotype in both models (compare panels A with B below), upregulating levels of both Rev-dependent unspliced (US or RRE) and Rev-independent multiply spliced (MS) transcripts as well as total HIV transcripts (TAR) as assessed by RT-qPCR. These findings were unexpected and suggested additional impact on viral transcription. Indeed, recent publication highlights that IGG and also Madrasin are rather poor splicing inhibitors as they exhibit direct effects on transcription (<https://doi.org/10.1371/journal.pone.0310519>). Moreover, our flow cytometry analysis in 6A12 clone (panel C) showed, in the absence of Rev, increases in the percentages of ECFP⁺ cells (lower dose of IGG) and appearance of mKO2⁺ECFP⁺ double positive cells (higher dose of IGG) which is in line with the qPCR results (panel A). Notably, appearance of mKO2⁺ECFP⁺ cells in the absence of Rev for higher dose suggest a spontaneous Rev-independent export upon IGG in the absence of Rev which could be explained by disturbing certain post-transcriptional RNA retention checkpoints. This puzzling phenotype will be studied in the future. However, at this stage we believe that interpretation of these results is complex and premature.

In addition to IGG, we also tested digoxin. However, due to high toxicity in J-Lat model of latency comparison studies with our model were precluded. Nevertheless, digoxin treatment in our 6A12 caused decreases in mKO2^{high+} cells and increases in the mKO2⁻ECFP⁺ population (panel D). This suggest that inhibition of splicing by digoxin causes decreases in the Rev-dependent and increases in Rev-independent expressions. It is again difficult to interpret this data at this stage and more studies are needed to validate our model for splicing by running sets of positive controls like knockdown of splicing factors and testing bigger panel of splicing inhibitors. This area of research is challenging due to co-transcriptional splicing events that could likely affect both stages of RNA biogenesis.

Consequently, we decided not to include our data using splicing modulators in the revised manuscript. We acknowledged that the loss of significant portions of the HIV-1 genome might be a limitation of our model, a point that we have now addressed in the revised manuscript (lines 408-412).

A**B****C****D**
The effect of depleting MATR3 using shRNA is consistent with previous reports but the magnitude of the effect raises questions regarding the utility of the system for more complex shRNA/CRISPR screens. It would have been interesting to demonstrate that the authors could enrich for shRNA sequences for MATR3 in a more complex library (i.e. 1 or 10% of the input library) as a demonstration of the suitability for such screens.

In this work, we used shRNA (shRNA-906) that was validated as the most potent (Sarracino et al. mBio 2018 <https://doi.org/10.1128/mBio.02158-18>). However, to elaborate on the magnitude on MATR3 knockdown on the Rev - dependent expression as also raised by the reviewer nr 1 (point 2 and point 4), we immunostained MATR3 and by using flow cytometry we addressed the correlation between MATR3 levels and % of mKO2-positive cells. As a control, we also analyzed the % of ECFP-positive cells. As shown in the new supplementary Figure 5B-C, we demonstrated a statistically significant correlation between MATR3 levels and %mKO2-positive cells and no correlation between MATR3 levels and %ECFP-positive cells. These results further validated the sensitivity of our model. We would like to also underscore that (as also highlighted by the reviewer 1, comment 2) MATR3 is likely a part of a broader network of cellular regulators (lanes 400-404) in which depletion of one factor might be compensated by function of other cofactors.

Major point

1) Contrary to the opening statement of the authors, I would consider HIV-1 Rev a well studied and well understood system. Multiple publications have explored the interaction between Rev and the RRE target RNA sequence, the cellular factors mediating the export of HIV-1 RNAs to the cytoplasm, and the composition of the exported RNA. The authors should acknowledge this point and appear to do in subsequent paragraphs.

We thank the reviewer for his/her valuable comment. We expanded the known Rev interactome as outlined in lines 56-85 and acknowledged that Rev is a well-studied and well understood system. We would like to point out that despite its extensive characterization, it is increasingly evident that additional host factors and regulatory mechanisms remain to be discovered. For instance, findings, including those implicating CRNKL1 and NF90ctv as negative regulators, highlight that the Rev system may be subject to more complex regulation than previously thought as also highlighted by us recently (DOI: 10.1016/j.jve.2024.100450). We have revised the introduction to better reflect both the established knowledge and the ongoing need for further exploration of this pathway.

2) Data in Fig. 3A and B appear inconsistent regarding the changes in mKO2 expression for clone 6A12. Flow shows are marked change in frequency of cells that are mKO2+ upon Rev addition in Fig. 3A but the MFI effect is only ~2 fold in Fig. 3B.

We thank the reviewer for pointing out this inconsistency between MFI for mKO2 and the frequency of mKO2-positive cells. Upon re-evaluating the flow cytometry analyses, we realized that we wrongly analyzed the MFI from “total cells population” instead of “ECFP-positive cells” as it was correctly performed for other figures in the submitted manuscript. We apologize for this mistake. Analyses from “total cell population” are indeed biased by transfection efficiency. By gating on ECFP+ cells we assure that Tat-transfected cells (ECFP is expressed under the control of LRT promoter) are analyzed. The new panel B from Fig. 3 has been updated. The description of the figure has been updated to clarify the gating (lane 494).

3) Blots in Fig. 5B regarding the changes in mKO2 expression do not appear to be consistent with the quantitation shown in Fig. 5C.

Thank you for noting the apparent inconsistency between the blots in Fig. 5B and the quantitation in Fig. 5C regarding mKO2 expression. As we conducted additional replicates of this experiment, we were able to select a more representative result for inclusion in the revised (new Figure 5D).

4) Supplementary Fig. 4. Western blots shown in Fig. 4B do not appear consistent with the quantitation in Fig. 4A. No double positive signal is shown in Fig. 4C with ABX464 addition but western blots show significant mKO2 expression in 4B.

Thank you for your comment. We replaced the Western blot with a more representative one (new Supplementary Figure 3D). Additionally, we included the analysis of double-positive populations to show the mKO2+ cells as requested by the reviewer (Supplementary Figure 3A).

Minor

point

1) The opening paragraph of the introduction, at a minimum, should provide references to the primary papers that have detailed the mechanism by which Rev regulates HIV-1 gene expression in support of the statements made.

Thank you very much for this very valuable comment. We introduced the pioneering studies of Malim/Cullen and Kijmes from early times of HIV research (Lines 46-51) and we apologize for our oversight in not doing so.

Re: Spectrum01903-24R1 (Development and characterization of a double-fluorescent HIV-1 reporter cellular model to tackle the Rev-dependent export pathway.)

Dear Prof. Anna Kula-Pacurar:

Your manuscript has been accepted, and I am forwarding it to the ASM production staff for publication. Your paper will first be checked to make sure all elements meet the technical requirements. ASM staff will contact you if anything needs to be revised before copyediting and production can begin. Otherwise, you will be notified when your proofs are ready to be viewed.

Sincerely,
Takamasa Ueno
Editor
Microbiology Spectrum